# New Quality-Range-Setting Method Based on Between- and Within-Batch Variability for Biosimilarity Assessment

**DOI:** 10.3390/ph14060527

**Published:** 2021-06-01

**Authors:** Alexis Oliva, Matías Llabrés

**Affiliations:** Departamento de Ingeniería Química y Tecnología Farmacéutica, Facultad de Farmacia, Universidad de La Laguna, 38200 Tenerife, Spain; mllabres@ull.edu.es

**Keywords:** analytical similarity, biosimilar, quality range method, between- and within-batch variability, bevacizumab

## Abstract

Analytical biosimilarity assessment relies on two implicit conditions. First, the analytical method must meet a set of requirements known as fit for intended use related to trueness and precision. Second, the manufacture of the reference drug product must be under statistical quality control; i.e., the between-batch variability is not larger than the expected within-batch variability. In addition, the quality range (QR) method is based on one sample per batch to avoid biased standard deviations in unbalanced studies. This, together with the small number of reference drug product batches, leads to highly variable QR bounds. In this paper, we propose to set the QR bounds from variance components estimated using a two-level nested linear model, accounting for between- and within-batch variances of the reference drug product. In this way, the standard deviation used to set QR is equal to the square root of the sum of between-batch variance plus the within-batch variance estimated by the maximum likelihood method. The process of this method, which we call QR_ML_, is as follows. First, the condition of statistical quality control of the manufacture process is tested. Second, confidence intervals for QR bounds lead to an analysis of the reliability of the biosimilarity assessment. Third, after analyzing the molecular weight and dimer content of seven batches of a commercial bevacizumab drug product, we concluded that the QR_ML_ method was more reliable than QR.

## 1. Introduction

During the last decade, developments in both technical and regulatory evaluation of biosimilars have led to the approval of >30 biosimilar monoclonal antibodies in Europe. The similar but not-identical concepts are now better defined, state-of-the-art analytical methods are available, and wider knowledge of product heterogeneities has been gathered from changes in postapproval manufacturing processes [1]. However, as manufacturers and regulators continue to gain experience and the number of approved biosimilars increases, ensuring consistency in the characteristics of these products over time emerges as a new regulatory and analytical challenge [2].

Characterization of biosimilar drugs involves a variety of analytical methods that permit comparing their physicochemical properties, biological activities, impurities, and stability. A wide range of state-of-the-art orthogonal methodologies are used to compare the physicochemical and biological properties of these types of products [3,4]. For example, peptide mapping with liquid chromatography tandem mass spectrometry (LC-MS/MS) is used for characterization of post-translational and chemical modifications. Gel electrophoresis and different chromatographic techniques are required for quality control and in batch-to-batch variability studies. The analysis of higher order structures is performed using techniques like proton nuclear magnetic resonance (1H NMR), circular dichroism (CD), fluorescence, Fourier transform infrared spectroscopy (FTIR), differential scanning calorimetry (DSC), and hydrogen–deuterium exchange MS (HDX-MS). In the case of monoclonal antibodies, in which function can influence the product safety and efficacy, an in-depth investigation of the mechanism of action (e.g., antibody-dependent cellular cytotoxicity and apoptosis) and additional complement-binding assays (e.g., Fc gamma receptor-binding activity) is required. Potency assay provides a quantitative measurement of biological activity, usually in clinical situations. This type of test is animal-based, cell-based, and uses biochemical assays.

Besides this, the consistency and robustness of the manufacturing process also needs to be demonstrated by implementing quality control, assurance procedures, and process validation. Although there are no specific types of assays for evaluating biopharmaceutical drugs, including biosimilar ones, selection of analyses is influenced by the properties of the reference product. However, the exact requirements can vary across regulatory agencies.

Xie et al. [5] published a list of analytical methods used for evaluating the similarity of a biosimilar drug, ranking relevant attributes according to risk. However, it is important to note that other methods may be appropriate for the biosimilar product under development, and future advances must be taken into consideration [4,6,7]. For this, the multiattribute method (MAM) is an emerging application of ultra-high-performance liquid chromatography, coupled with mass spectrometry, useful for simultaneous monitoring of multiple attributes of biopharmaceutical product quality [8].

Various biological reference preparations, each with a distinct role, are used during development and manufacture of a biosimilar product. The proprietary manufacturer’s reference standards for bioassays are used to support the characterization and traceability of the product’s critical quality attributes throughout its life cycle, both horizontally (between products and batches) and longitudinally (over extended periods). Lee at al. [9] used internal reference standards obtained from a commercial-scale run in the similarity assessment of an adalimumab biosimilar.

Recent studies have shown shifts in the critical quality attributes (CQAs) of batches of biosimilar rituximab and trastuzumab products, including bioactivity [9], associated with postapproval manufacturing changes. In such a situation, regulatory authorities can assess the shifts using “in-house” reference standards, although the availability of international standards (ISs) would allow them to identify the causes of variation and determine their clinical significance [10]. For this, regulatory guidelines request that manufacturers monitor and control the CQAs of biosimilar drugs to keep them within appropriate ranges, so that their quality and clinical properties remain consistent over time [11]. Assessment of analytical similarity requires the comparison of different batches to describe and analyze the variability between them.

In September 2017, the FDA published guidance on statistical approaches for analytical similarity assessment [12]. It recommended a stepwise approach by tier-based quality-attribute evaluation. Equivalence testing is applied for attributes ranked as tier 1, those with the highest potential clinical impact. Due to its problems in fixing the similarity margin, various authors have criticized the equivalence test in tier 1 [13,14]. However, the FDA withdrew this guidance in June 2018 due to industry comments [15]. In May 2019, it published another guidance, *Development of Therapeutic Protein Biosimilars: Comparative Analytical Assessment and Other Quality-Related Considerations* [16]. In this new guidance, the FDA proposed the use of comparative analytical assessment by the quality range (QR) approach, under the assumption that the biosimilar product and the reference product have similar population means and standard deviations [13]. Nevertheless, various authors have questioned this approach [17,18,19,20].

Both equivalence tests in tier 1 and QR approaches present the following characteristics: each lot contributes one test value to the CQA being assessed, to avoid bias in the estimation of residual variance [13]. Wang and Chow [21] proposed an alternative approach to test multiple samples from each lot, in order to account for the worst possible scenarios for fair and reliable comparisons. For this, they took the within-lot and between-lot variabilities into consideration for the analytic similarity assessment.

In addition, the QR method ignores: (1) lot-to-lot variability, (2) the differences or shift between means, and (3) relative variability between the reference product and the biosimilar product. These limitations are the most criticized issues, and could have a large impact on the use of the QR method for analytical similarity assessments. However, the equivalence test presents the same limitations [22]. Many authors have studied and discussed these aspects from various points of view, proposing a range of equivalence tests.

The two modified versions of the QR method proposed by Son et al. [17] not only overcome the limitations of the QR method mentioned above, but can help in detecting product changes during the manufacturing process. However, in these approaches, the within-lot variability was not considered. Oliva and Llabrés [23] applied the QR method for analytical similarity evaluation of various bevacizumab lots. In this study, the two main sources of variability deriving from the estimates of the standard deviation of the population of reference products lots (σ_R_) were analyzed. The within-lot variation (i.e., the analytical method uncertainty) was known and in control through the analytical method validation. However, the between-lot variability of the reference product required greater vigilance and monitoring by the manufacturer. The presence of a lot with unexpected content may result in rejection by the biosimilarity test.

Recently, Oliva and Llabrés [24] analyzed the limitations of the QR approach in analytical similarity assessments. There are two kinds of shift: in means and in variability. However, the problem is (i) how to detect and quantify the shift, and (ii) how it affects product safety and efficacy. Only when the product is in control; i.e., within certain in-house specification limits, can it be released for use [25]. For this, the QR approach may be an option.

This study presents an estimation of QR bounds based on the variance components to account for both between-lot and within-lot variability; variance components were computed by the maximum likelihood method using a linear random model. We call this method $QR_{ML}$ to differentiate it from the currently used procedure based on one sample per batch. For this, the molecular weight (Mw) and dimer content (expressed as percentage) were used as CQAs. We used real data from various bevacizumab commercial lots manufactured at different times, which showed at least 1-year residual shelf life.

## 2. Results and Discussion

### 2.1. Analytical Methods

The Mw of the intact molecule is the first physical parameter to be analyzed since it defines the identity, its derivatives and confirm the primary structure. From instrumental point of view, the MS is the first option. However, size-exclusion chromatography (SEC) is the common method used to characterize protein aggregates. It enables the separation of the monomeric protein and derivatives on the basis of their hydrodynamic radius, simple to use and compatible with high-throughput mode. This technique combined with detection modes different such as refractive index (RI), light-scattering (LS) and viscometry (VM) allows comprehensive characterization of proteins [26]. For this, the Mw and dimer, expressed as percentage, were used as CQAs. The first, provide information about the identity of the protein and the second, the content of impurities.

The SEC coupled with RI detector (SEC/RI) system was used for estimating monomer and dimer content through relative peak area percentage values. The SEC/LS system was used in the characterization process (i.e., identity of starting proteins, products and intermediates of protein aggregation). It enables the Mw and size distribution to be determined.

#### In-Study Validation

The two proposed analytical methods were validated using the in-study validation procedure. In all these procedures, it was necessary to consider the measurement uncertainty associated with the results and to determine whether the level of variability satisfied the specification limits [27].

For this, the X-bar and moving range (MR) control charts were used to verify that the method remained stable and in control over time. In this case, quality-control samples were analyzed each working day to monitor the method parameters; the process was carried out during a two-year period. The X-bar and MR control charts showed that the SEC/LS method was in control (i.e., within acceptable bounds) and stable. Assuming that Mw data were normally distributed, the mean Mw was estimated to be 148.9 kDa from the X chart, whereas the process variation was estimated to be 1.06 kDa from the MR chart. The overall uncertainty, calculated as the sum of the uncertainty of each component’s contribution (precision and accuracy), was 0.98 kDa. The expanded uncertainty was 1.96 kDa, using a coverage factor of 2. The calculated Mw was thus 148.9 ± 1.96 kDa [28]. In a previous work, the overall uncertainty calculated using the Bayesian posterior distribution [29] was lower than 2%. In addition, the reliability of this analytical method was also assessed by analyzing eight standard proteins in the range of 14.4 to 443 kDa. The average error between the observed and predicted Mw ranged from 1.32% to 6.90%, with an overall mean average error of 2.8% for the proteins examined. The results showed an intra- and interassay precision, expressed as relative standard deviation, lower than 3.0% [26].

The SEC/RI method was also in control and stable. The dimer mean percentage was estimated to be 1.569% from the X chart (*n* = 51). At first, all individual measurements were within the lower (LCL) and upper (UCL) control limits (1.089, 2.048) calculated according to Montgomery [30]. The MR chart allowed the method’s standard deviation to be estimated at 0.16% [23].

### 2.2. Statistical Model

The statistical model used was:(1)$$y_{ij}=\mu +B_{i}+\epsilon_{ij}$$
where $y_{ij}$ is the observation *j* (*j* = 1, ..., *n_i_*) from batch *i* (*i* = 1, ..., *m*), *µ* is the general mean, *B_i_* is the batch random effect, and $\epsilon_{ij}$ is the residual random term accounting for sampling variability and analytical method uncertainty. Random terms $B_{i}$ and $\epsilon_{ij}$ are assumed independent and with distribution $N\left(0\;,\;\sigma_{B}^{2}\right)$ and $N\left(0\;,\;\sigma^{2}\right)$, respectively. The intraclass correlation coefficient was:(2)$$\rho =\frac{\sigma_{B}^{2}}{\sigma_{B}^{2}+\sigma^{2}}$$

In this study, we analyzed the QR interval computed from the maximum likelihood estimates, usually defined as:(3)$$QR_{ML}=\left[\hat{\mu }-k\;{\hat{\sigma }}_{R}\;,\hat{\mu }+k\;{\hat{\sigma }}_{R}\;\right]$$
where $\bar{\mu }$ is the sample mean; the multiplier *k* is a constant, which in the QR method is usually 2 or 3 based on risk analysis; and ${\hat{\sigma }}_{R}$ is the estimated standard deviation for the reference medicinal product, calculated from the variance components of the model (Equation (1)):(4)$${\hat{\sigma }}_{R}=\sqrt{{\hat{\sigma }}_{B}^{2}+{\hat{\sigma }}^{2}}$$

Variance components of the statistical model (Equation (1)) can be estimated using least square method (i.e., ANOVA), but only if the experimental design is balanced (same number of observations per batch). However, even in these circumstances, there is some probability of a negative ${\hat{\sigma }}_{B}^{2}$ when $\sigma_{B}^{2}\ll \;\sigma^{2}$. Therefore, we chose to estimate the variance components of the statistical model by the maximum likelihood method. This was done using the function lmer, included in the library lme4 [31] of the R software [32]. Confidence intervals for the bounds of $QR_{ML}$ are computed using the function bootMer() included in the same package (see Appendix A).

### 2.3. Data Analysis

Figure 1 depicts individual observations for each batch, together with $QR_{ML}$ estimates for *k* = 2 (inner lines) and *k* = 3 (outer lines). As would be expected, *k* must be equal to 3 to assure that all observations will be inside QR limits with a high probability. Table 1 shows the summarized statistics: number of observations per batch, with mean and standard deviation for both quality attributes included in this study. The overall bevazicumab mean molecular weight was 148.79 kDa; the within-batch standard deviation ranged from 0.294 to 2.07 (coefficient of variation from 0.26% to 1.38%). The overall bevacizumab mean dimer content was 1.539%; the within-batch standard deviation ranged from 0.034 to 0.198 (coefficient of variation from 2.30% to 12.7%).

Table 2 shows the variance components estimated by the maximum likelihood method with their 95% confidence intervals. As we concluded from these results, we were required to accept the null hypothesis $H_{0}:\sigma_{B}^{2}=0$ for both molecular weight and percent dimer content because the confidence intervals included the zero value. Therefore, when at least taking the precision of the analytical methods into account, there were no differences between batches, and the observed variability was due to within-batch factors. These were mainly analytical method uncertainty and sampling variability.

Table 3 shows the QR intervals and 95% confidence intervals for both lower and upper limits. For molecular weight, the estimated QR intervals were (145.21, 152.40), and the 95% confidence intervals were (144.33, 146.13) for the lower bound and (151.50, 153.27) for the upper bound. For dimer content (%), the estimated QR intervals were (1.140, 1.938), and the 95% confidence intervals were (1.055, 1.225) for the lower bound and (1.854, 2.026) for the upper bound. To show that the $QR_{ML}$ method provided more reliable results than those given by the one in use to date (calculation of QR taking one sample per batch), we computed the expected distribution of the latter using a stratified bootstrap sampling (one sample per batch; 2000 simulations) from the experimental data. Figure 2 shows the histograms for the lower and upper values of the QR for the quality attributes of molecular weight (top left and middle) and dimer content (bottom left and middle). The vertical lines correspond to the values estimated for $QR_{ML}$ (see Table 3). We have also included the plots of the upper vs. lower bounds of the computed QR ranges (top and bottom right). These graphs clearly show that the QR bounds were negatively correlated.

Classical application of statistical process control implies that when the manufacturing process is under statistical control, only “natural variability” operates, and therefore batch-to-batch variability is explained solely by within-batch variability. Under these circumstances, an easy way to estimate the standard deviation for the reference product ($\sigma_{R}$) would be from combined variance of several samples from several batches. However, manufacture of biopharmaceuticals is a complex process, and due to the lack of published manufacturing records, we could not assure that batch-to-batch variability was negligible. The importance of within-batch variability can be better generalized and analyzed if we express $QR_{ML}$ (Equations (2) and (5)) as a function of the intraclass correlation coefficient (Equation (2)):(5)$$QR_{ML}=\left[\mu -k\frac{\sigma_{B}}{\sqrt{\rho }}\;,\mu +k\;\frac{\sigma_{B}}{\sqrt{\rho }}\;\;\right]$$

Table 4 shows QR values for several combinations of $\sigma_{B}$ and ρ using Equation (5), μ = 100 and *k* = 3, where the relevance of the intraclass correlation coefficient is seen.

The relevance of the uncertainty of the analytical method can be anticipated from a proper validation procedure. Uncertainty regarding the molecular weight determination already published [26,28] was of the same order of magnitude as the value estimated for σ (see Table 2). However, we have found some discrepancies in the determination of dimer content. The value obtained in this study (0.133; see Table 2) was larger than expected from the validation of the analytical method. This was probably because aggregation can take place after filling vials, leading to a larger sample-to-sample variability.

## 3. Materials and Methods

### 3.1. Reference Medicinal Product

The Avastin^®^ (bevacizumab) was produced by Genentech Inc. (Roche labs, UK) and supplied as a 25 mg mL^−1^ solution in phosphate buffer (pH = 6.2), containing 159 mM trehalose dehydrate and 0.04% *w*/*v* polysorbate 20. Deionized water was purified in a MilliQ Plus system from Millipore (Molsheim, France); all other chemicals and reagents were HPLC grade. All solvents were filtered with 0.45 µm (pore size) filters (Millipore) and degassed.

### 3.2. Chromatography System

#### 3.2.1. SEC/RI System

The chromatographic system used was a Waters apparatus (Milford, MA, USA) consisting of a pump (600E Multisolvent Delivery System), an auto sampler (700 Wisp model) and a differential refractive index (RI) detector (Waters 2414 model). Elution was performed at room temperature in a Protein KW-804 column (8 mm × 300 mm, Waters), and the mobile phase was phosphate-buffered saline (300 mM NaCl, 25 mM disodium hydrogen phosphate, pH 7.0) at a flow rate of 1.0 mL min^−1^, and injection volume 25 µL. The data was collected and analyzed using the Millennium 32^®^ chromatography program (Waters). This software was used for chromatogram integration and estimating monomer and dimer content through relative peak area percentage values.

#### 3.2.2. SEC/LS System

The light-scattering (LS) multiangle miniDawn detector (Wyatt Technology) was placed downstream of the column and upstream of the RI detector. To reduce baseline noise, a pulse dampener (Alltech Associates, Deerfield, IL, USA) was connected downstream of the pump, and two 25 mm high-pressure filters (Millipore) with 0.22 and 0.1 µm pores, respectively, were used for online filtration of the mobile phase. The column and other chromatographic conditions were identical to those used for the SEC system mentioned above. A 100 µL sample of each solution was injected into the system, and data collection and Mw calculations were performed using ASTRA software, version 6.1.0 (Wyatt Technology), using the Debye fitting method. The dn/dc value was determined by analyzing the concentration dependence of the bevacizumab refractive index, using a differential refractometer (Waters 2414) with the mobile phase as solvent. The estimated dn/dc was 0.184 ± 0.003 mL g^−1^ (*n* = 10) [33].

## 4. Conclusions

In short, the method proposed in this paper showed several advantages over the traditional one. First, variance components from the statistical model allowed the type of variability of the reference product to be analyzed, namely the between-batch variability and within-batch variability; the latter accounting mainly for the analytical uncertainty and sampling randomness. Second, interpretation of the uncertainty of the analytical model led us to the biosimilarity design. Third, it facilitated the calculation of more precise QR values. Fourth, it was suitable for setting a priori values for QR.

## Figures and Tables

**Figure 1 pharmaceuticals-14-00527-f001:**
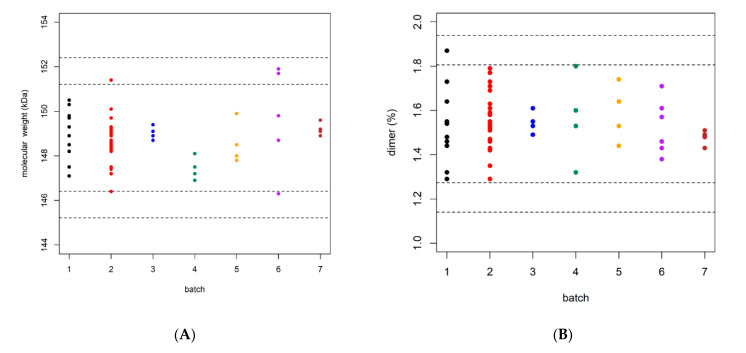
Data sets and QR estimated from variance components for *k* = 2 (inner lines) and *k* = 3 (outer lines). (**A**) correspond to molecular weight and (**B**) for dimer CQAs.

**Figure 2 pharmaceuticals-14-00527-f002:**
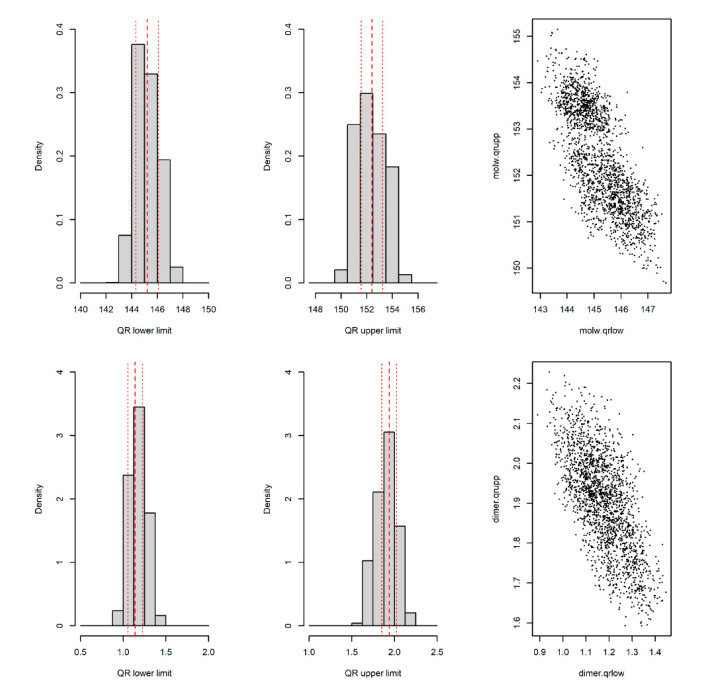
QR bounds distributions of stratified bootstrap samples and estimates with 95% confidence intervals for QR estimated from variance components.

**Table 1 pharmaceuticals-14-00527-t001:** Summarized statistics of experimental data (sample size, mean, and standard deviation) for molecular weight (Mw) and dimer content as percent of total bevacizumab concentration (dimer).

**Batch**		1	2	3	4	5	6	7
**Sample Size**		10	23	4	4	4	6	4
**Mw (kDa)**	**Mean**	148.98	148.64	149.03	147.43	148.55	149.70	149.20
	**SD**	1.151	1.014	0.299	0.512	0.947	2.070	0.294
**Dimer (%)**	**Mean**	1.532	1.544	1.545	1.562	1.587	1.527	1.478
	**SD**	0.178	0.132	0.050	0.198	0.130	0.125	0.034

**Table 2 pharmaceuticals-14-00527-t002:** Maximum likelihood method estimates and their corresponding 95% confidence intervals.

CQA	Parameter	Estimation	Lower Bound	Upper Bound
Mw (kDa)	${\hat{\sigma }}_{B}$	0.403	0.000	1.010
	$\hat{\sigma }$	1.128	0.936	1.416
	$\hat{\mu }$	148.81	148.32	149.29
Dimer (%)	${\hat{\sigma }}_{B}$	0.000	0.000	0.048
	$\hat{\sigma }$	0.1330	0.1106	0.1609
	$\hat{\mu }$	1.539	1.504	1.5750

**Table 3 pharmaceuticals-14-00527-t003:** Estimates and bootstrap 95% confidence intervals for QR estimated from variance components.

CQA	QR Bounds	Lower	Estimate	Upper
Mw (kDa)	Lower	144.33	145.21	146.13
	Upper	151.50	152.40	153.27
Dimer (%)	Lower	1.055	1.140	1.225
	Upper	1.854	1.938	2.026

**Table 4 pharmaceuticals-14-00527-t004:** QR values as a function of σ_R_ and ρ for *μ* = 100 and *k* = 3.

$\sigma_{B}$	ρ	QR	
		Lower	Upper
1.25	0.1	88.1	118.6
	0.3	93.2	106.8
	0.5	94.7	105.3
2.50	0.1	76.3	123.7
	0.3	86.3	113.7
	0.5	89.4	110.6
5.00	0.1	52.6	147.4
	0.3	72.6	127.4
	0.5	78.8	121.2

## Data Availability

Data are available within this article and in the associated Appendix A.

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
