# Peer review of "New Quality-Range-Setting Method Based on Between- and Within-Batch Variability for Biosimilarity Assessment"

_pharmaceuticals, 2021, doi:10.3390/ph14060527_

Round 1

Reviewer 1 Report

The authors have presented a manuscript debating the importance of the biosimilarity assessment of biosimilars.

The topic itself is important, and as the authors stated in the introduction - this is still a developing field, with even the FDA guidelines changing, adapting and developing.

There are a few suggestions to improve the manuscript:

1) Title: the title should be changed. in its present form it looks more like a review paper, and the fact that the authors are proposing a new method/approach is not visible. The fact that the method was applied to bevacizumab could also be mentioned.

2) abstract: again the novelty of the proposed method is not stated, whereas in the conclusion of the paper authors stated "the method proposed in this paper...."

This should be visible also in the title and in the abstract

3) Materials and Methods: please include the full name and then introduce the abbreviation of the LS detector.

There are many references introduced in the Materials&Methods section. This is not very common. Please comment why this is necessary.

4) english improvement: the text should be checked once again. Having it edited by a native english speaker would improve it significantly.

There are a few examples where due to some english grammar and spelling mistakes, the complete meaning of the sentence is missed.

Examples: abstract "it is responsibility of the promoter their selection"; introduction page 3 line 91 "with different proposed for equivalenced test"; introduction page 3 line 107 "how to affect the product safe and efficacy"

Additional typos: page 2 line 30: is/are; page 4 line 13 "together their 95% confidence intervals"  - is "with" missing?; page 4 line 13 "there is not differences" please change to NO; page 9 line 24 "the second method are also in control...." please change to IS.

Author Response

REPLIES TO REVIEWERS´ COMMENTS

Reviewer #1

Points 1

Title: the title should be changed. In its present form it looks more like a review paper and the fact   that the authors are proposing a new method/approach is not visible. The fact that the method was applied to bevacizumab could also be mentioned.

Point 2.

Abstract again the novelty of the proposed method is not stated, whereas in the conclusion of the paper authors stated “the method proposed in this paper…”

This should be visible also in the title and in the abstract.

 Thank for your suggestion. The title and abstract were changed.

 Point 3. Materials and Methods, please include the full name and then introduce the abbreviation of the LS detector.

This change was made.

There are many references introduced in the Materials and Methods section. This is not very common. Please comment why this is necessary.

 It is true, there are many references in this section. However, it is necessary to include it since all references are related with the validation of different analytical methods used in our work. In addition, the uncertainty of the analytical method and sampling variability play a vital role in the proposed method for similarity assessment.

Point 4. English improvement.

The text was carefully revised by a native speaker.

Additional types.

All them were revised and changed.

Reviewer 2 Report

The authors compare two methods for measuring similarity. The approach is interesting and the results clearly presented. The pictures are low resolution and need to be upgraded. Furthermore there are errors in the references. Please correct them and update the referencing style.

Author Response

Reviewer #2:

The authors compare two methods for measuring similarity. The approach is interesting and the results clearly presented.

Thank you very much for yours comments about our paper.

The pictures are low resolution and need to be upgraded.

The figures were improved, especially, resolution (600)

Furthermore, there are errors in the references. Please correct them and update the referencing style.

It is true, there are many errors in the references. All references were carefully revised following the journal referencing style.

Reviewer 3 Report

I suggest to improve method description. i recommend the publication.

Author Response

Reviewer #3

 I suggest to improve method description. I recommend the publication.

 We think that the method description is appropriate from statistical point of view, although it is difficult to understand the application of our method, especially, the equation #1 (in the original version) since it is necessary to read before the 3.4 section in “Material and methods”. For this, we have moved this section to “Results and discussion” called “Statistical model” in order to facilitate the read and the data analysis.